# Characterization of Mixed Pellets Made from Rubberwood (*Hevea brasiliensis*) and Refuse-Derived Fuel (RDF) Waste as Pellet Fuel

**DOI:** 10.3390/ma15093093

**Published:** 2022-04-25

**Authors:** Rattikal Laosena, Arkom Palamanit, Montri Luengchavanon, Jitralada Kittijaruwattana, Charoen Nakason, Seng Hua Lee, Aujchariya Chotikhun

**Affiliations:** 1Energy Technology Program, Faculty of Engineering, Prince of Songkla University, Hat Yai 90110, Songkhla, Thailand; 5840310525@psu.ac.th; 2Energy Technology Program, Department of Specialized Engineering, Faculty of Engineering, Prince of Songkla University, Hat Yai 90110, Songkhla, Thailand; arkom.p@psu.ac.th; 3Center of Excellence in Metal and Materials Engineering (CEMME), Sustainable Energy Management Program, Wind Energy and Energy Storage Centre (WEESYC), Faculty of Environmental Management, Prince of Songkla University, Hat Yai 90110, Songkhla, Thailand; montri.su@psu.ac.th; 4Faculty of Science and Industrial Technology, Prince of Songkla University, Surat Thani Campus, Mueang, Surat Thani 84000, Surat Thani, Thailand; jitralada.k@psu.ac.th (J.K.); charoen.na@psu.ac.th (C.N.); 5Institute of Tropical Forestry and Forest Products, Universiti Putra Malaysia, Serdang 43400, Selangor, Malaysia; lee_seng@upm.edu.my

**Keywords:** wood pellet, rubberwood, refuse-derived fuel, energy potential, pellet fuel

## Abstract

The objective of this study was to investigate the production and properties of mixed pellets made from rubberwood (*Hevea brasiliensis* Muell. Arg) and refuse-derived fuel (RDF) waste with no added binder. Three different conditions of mixed pellets were developed to compare their chemical and physical properties to rubberwood pellets. The produced samples were subjected to both ultimate and proximate analyses. The contents of C, H, N, S, and Cl significantly increased with the increasing amount of refuse-derived fuel in the samples, resulting in reduction of the volatile matter. The mechanical durability of the pellet samples ranged between an average value of 98 and 99%. Mixed pellets containing 50% of rubberwood and 50% of refuse-derived fuel have improved heating values by 22.21% compared to rubberwood pellets. Moreover, mixed pellets having 50% of wood and 50% of refuse-derived fuel had the highest density and the highest energy compared to the other samples. Based on the findings of this study, it appears that the manufactured mixed pellets have the potential to be used as high-energy fuel.

## 1. Introduction

Wood pellets are typically made from compacted sawdust or other woody materials, meaning that they are related to biowaste utilization [1]. Pellets produced by the compression process are uniform in size and shape, have a high density, and contain little moisture and ash [2]. Wood pellets have the advantage of being easier to store and transport over long distances than wood chips. Furthermore, fresh wood has a calorific value of 9–12 GJ/ton, whereas wood pellets have a calorific value of around 16–18 GJ/ton [3]. Currently, the demand for wood pellets in the European Union, which produces roughly half of the world’s wood pellets, is expected to grow by more than 22.5 million tons [4].

Rubberwood (*Hevea brasiliensis* Muell. Arg) is the most vital raw material for the wood-based industries in both Thailand and Malaysia [5]. Rubberwood is primarily used in the home furniture industry as a raw material. Following primary processing of the wood, the remainder of the raw material becomes wood residues, such as shavings, slabs, and sawdust [6]. According to Ratnasingam et al. [7], for 3.1 million m^3^ input of rubberwood logs to sawmills, 3944 m^3^ sawdust could be generated during the sawmilling process. It was noted that this sawdust was used by the sawmills for fuel generation. Rubberwood waste is abundant in Thailand’s southern provinces, and it has great potential for biofuel applications [8]. Rubberwood is a plentiful, renewable, and long-lasting resource that is used to make furniture and wood-based panels. In Thailand, the wood residues from these products have a lot of potential for bioenergy applications [9]. Biomass pellets could be made from logging waste and industrial waste from rubberwood-based products [10,11]. Thailand also increased wood pellet production in 2019 for both export and domestic use, exporting 172,441 tons of wood pellets [1].

Rubberwood is known for its great potential as a source of energy. In comparison to other types of biomasses, rubberwood displayed higher potential energy production compared to that of empty fruit bunches and palm kernel shells [12]. Comparing to pellets made from oil palm biomass, which is widely available in both Malaysia and Thailand, rubberwood exhibited higher potential in pellet production, owing to its higher calorific value than palm kernel shell and fiber. In addition, pellets made from rubberwood also offer several advantages: (i) they have better and more uniform heating properties per unit volume; (ii) they generate fewer particulate emissions during burning; and (iii) they have a lower transportation fee, due to their increased bulk density [12].

Refuse-derived fuel (RDF), on the other hand, is a by-product of municipal solid waste (MSW), which is expected to reach 2.6 billion metric tons by 2030 [13]. MSW landfills are usually made up of 50–60% soil-type material, 20–30% combustible components, 10% inorganic components, and a small amount of metals [14]. Open dumping was the most common MSW disposal method in developing countries over the previous decade, with solid wastes at dumpsites being used as refuse-derived fuel (RDF) [15]. RDF is mostly made up of carbon-based derivatives, such as organics, plastic, paper, wood, and textiles, with plastic and paper accounting for 50–80% of RDF composition [16]. Pelletization could be used to give RDF a uniform shape and density, allowing it to be used as a fuel. However, one of the drawbacks of RDF pelletization is the low lignin content. Several woody biomasses could be mixed with RDF for pellet production as a viable solution in this case.

Several types of vegetal biomass materials have been co-pelletized with MSW for pellet production. Wood and energy crop residues with different mix compositions has been used to improve the quality of wood pellets while lowering their cost [17]. As a result, many researchers have concentrated on using co-energy from wood and non-wood as a sustainable alternative fuel [18]. Adding 5% binder concentration to RDF pellets made from MSW and rice husk, for example, increased the calorific value of pellet samples [19]. The use of durian waste (shell and seed) as a feedstock in fast pyrolysis to produce RDF pellets was investigated [20]. Furthermore, Cui et al. [21] discovered that co-pelletizing of biomass and waste appears to be a promising way to improve the competitiveness of biomass pellet fuel manufacturing at scale in the future. In previous research studies, it was found that co-pyrolysis of wood pellet and polyethylene increases gas yield, due to oxygenates and moisture in the wood pellet [22]. In the last decade, researchers have also investigated the pyrolysis mechanisms of biomass and plastics [22,23,24,25]. Kumagai et al. [26] investigated the co-pyrolysis interactions of beech wood and polyethylene (PE). The findings revealed an understanding of the operative mechanism in the co-pyrolysis of wood materials and synthetic polymers, and this could be useful in the future for pyrolysis reaction control and recovering desirable products from difficult-to-separate initial mixtures.

To the best of the authors’ knowledge, there have been few studies on the co-pelletization of rubberwood and RDF. As a result, knowledge of the effects of co-pelletization of rubberwood and RDF on the performance of pellets produced is limited. Therefore, the objective of this research was to investigate the energy potential and some properties of mixed pellets made from rubberwood and RDF. Furthermore, the results of this research could provide evidence that mixed pellet fuel can be used as a feed stock in thermal waste-to-energy technologies.

## 2. Materials and Methods

### 2.1. Pelletization and Sample Preparation

Commercially produced rubberwood sawdust, a waste product supplied by BNS Wood Industry Co., Ltd., a sawmill located at Mueang, Surat Thani, Thailand, was dried in a hot-air oven at a temperature of 103 ± 2 °C for 48 h to reach a moisture content of 0%. Before being used to make pellet samples, dried sawdust was screened on a sieve with 18 mesh to remove any oversize particles. Municipal solid waste (MSW) samples that had been landfilled for 1.5–2 years were collected from a landfill area in Pattalung Province, Thailand. The collected MSW samples were separated into combustible and incombustible materials, using a separating machine and manual selection by workers. After separation, the combustible materials were converted into refuse-derived fuel (RDF) by a machine operated by a private company in Thailand’s Mueang, Nakhon Si Thammarat. 

RDF wastes that have been processed to separate glass, metal, and inorganic materials were shredded to the point where 95% of the weight passes through a 2 in. square mesh screen and was denoted as RDF-3 [27]. Finally, raw RDF-3 was shredded to a size suitable for pelletization, as illustrated in Figure 1.

Table 1 displays the percentage composition of materials by weight for four different types of pellets. Rubberwood sawdust and shredded RDF-3 were thoroughly mixed in a mixing drum for 5 min. The moisture content of the mixtures was around 14–16%. Following that, the mixed materials were pelleted in an electric flat die wood pellet mill, KN-D-200, with 7.5 hp (380v), 50 Hz (Figure 2), and a pellet mill die in 6 mm to produce 5 kg of each condition. The pellets were air-dried and cooled before being stored in a conditioned room with a temperature of 25 °C and a relative humidity of 65%.

### 2.2. Properties Evaluation of Pellets

#### Physical Properties of the Samples

The dimension, density, and color of the samples were determined. Three replications of each pellet type were completely randomized and used for sample testing. 

The samples’ dimensions and weights were measured and weighed with precisions of 0.01 mm and 0.0001 g, respectively. The density of the pellet was calculated by using the following equation: D = M/V(1)
where D is the density of pellet (g/cm^3^), M is the mass of the pellet (g), and V is the volume of pellet (cm^3^).

### 2.3. Mechanical Durability of the Samples

Mechanical durability testing of the samples was carried out in accordance with the procedure outlined in EN 15210-1. It is a 10 min tumbling box test that determines the resistance of densified fuels to shocks and/or abrasion caused by handling and transportation processes (EN 15210-1: 2009). The broken pellet pieces and dust were separated and weighed by using a 3.15 mm sieve. The mechanical durability was calculated by using the following equation: DU = (M_A_/M_B_) × 100(2)
where DU is the mechanical durability (%), M_A_ is the mass of the pellet pieces after tumbling (g), and M_B_ is the mass of the pellet pieces before tumbling (g).

### 2.4. Calorific Value of the Samples

The heating value or gross calorific value of pellet samples was determined by using an automatic bomb calorimeter, Leco A-350, St. Joseph, MI, USA. The measurements were made in triplicate, and the results are given as means with standard deviations in MJ/kg (ASTM D 3286-96). The samples were also subjected to proximate analysis in order to determine the impact of moisture content (MC), volatile matter (VM), ash content (AC), and fixed carbon (FC) on the gross calorific value. The VM, AC, and FC quantities were determined by measuring the mass loss that a sample undergoes at a temperature of 900 °C, under a nitrogen atmosphere, and then held at 900 °C, under atmospheric air [28].

### 2.5. Ultimate Analysis of the Samples

The samples were finally analyzed by using the ASTM D5373-93 (1997) procedure, ASTM, West Conshohocken, PA, USA. In an ultimate analysis, the contents of elemental carbon (C), hydrogen (H), nitrogen (N), and sulfur (S) in pellets were determined by using a Perkin Elmer, 2400 Series II CHNS/O analyzer, USA. The chlorine (Cl) content was determined by using a 785 DMP Titrino from Metrohm in Switzerland.

### 2.6. Microstructure Evaluation of the Samples

A scanning electron microscope (SEM), FEI Quanta 250, Waltham, MA, USA was also employed to examine the microstructure samples. The images were captured by using a SEM set to 15 kV, and all of the specimens were coated with a thin gold layer prior to analysis. The images were taken from cross-sections of pellets with a diameter of 6 mm.

### 2.7. Data Analysis

For data analysis, a completely randomized design of sample types was used. Using XLSTAT in Microsoft Excel 365^®^, Microsoft, Redmond, WA, USA, analysis of variance (ANOVA) was used to determine the significant differences between the four types of pellet specimens. Duncan’s multiple range tests, as well as SPSS Statistics version 22, were used for additional analysis. A *p*-value of 0.05 was used as the level of confidence.

## 3. Results and Discussion

Table 2 summarizes the physical properties of the pellet samples, such as the diameter, length, density, and color appearance.

Figure 3 depicts two types of pellets that revealed the color comparison between pure and mixed wood pellets. Rubberwood pellets are brown in color, but they turned black when RDF-3 was mixed with rubberwood during pellet production. It was confirmed that the color of the wood pellets could be altered based on their material composition. The diameter of the pellets ranged from 6.11 to 6.21 mm. As the pellets were produced by using a flat die mill of 6 mm, the diameters did not differ much within different types of pellet formulations. Generally, the pellets made by mixing rubberwood and RDF-3 have slightly larger diameters compared to those of the pellets made from pure rubberwood. In terms of length, the pellets made from mixing 70% rubberwood and 30% RDF-3 (W7R3) have the longest length, of 43.40 mm. The length decreased when higher loading of RDF-3 was added, but they were still longer than the pellets made with pure rubberwood (36.27 mm). Meanwhile, the density of the pellets made with pure rubberwood was the highest (1.288 g/cm^3^) when compared to the pellets with the addition of RDF-3.

Table 3 displays the mechanical durability and heating value of pellets of various pellet formulations. The mechanical durability of the pellets manufactured in this study ranged from 98.27 to 99.07%. The results of this study indicate excellent mechanical durability, which is consistent with the findings of Ungureanu et al. [29], who reported values of 96–97%. All samples with a durability greater than 96% were considered to be of high quality, according to biomass pellets standards [30]. This result demonstrated that both materials can be successfully blended to produce pellets without the addition of any adhesive. The addition of 40% RDF-3 and higher appears to improve the mechanical durability of the pellets. Pellets made with 40% and 50% RDF-3 loadings have significantly higher mechanical durability than pellets made with pure rubberwood. The improvement could be related to the density of the pellets, as higher specific densities are generally associated with greater durability [30]. Pellets with a 50:50 rubberwood:RDF-3 ratio have the highest density, and, thus, better mechanical durability is anticipated. 

As shown in Figure 4, pure rubberwood sawdust pellets had an average calorific value of 17,277 MJ/kg, while mixed pellets of W7R3, W6R4, and W5R5 had 18,866, 19,461, and 21,445 MJ/kg, respectively. Previous research discovered that the calorific value of RDF in Latvia and Lithuania was 18,310–22,521 MJ/kg [31]. The heating value result showed that mixing RDF-3 can increase the energy of pellet (*p* < 0.01). Furthermore, when the RDF-3 ratio is increased, the calorific value improves in a linear fashion. It is a fact that this value is dependent on the MSW source, which contains a variety of waste compositions. Rezaei et al. [32] investigated the heating value of pellets made from various plastic, paper, organic, and wood compositions. Pellets with the highest plastic and lowest paper content were found to have the highest heating value. It was reported that pellets with the highest plastic and minimum paper contents generated the highest heating value.

An ultimate analysis was carried out to investigate the carbon, hydrogen, nitrogen, sulfur, and chloride content of the pellets. The results are listed in Table 4. Generally, increased contents of carbon, hydrogen, nitrogen, sulfur, and chloride were observed when higher ratios of RDF-3 were added. Pellets made with 50% RDF-3 addition have the highest carbon (50.78%), nitrogen (0.69%), and chloride (0.124%) contents. Meanwhile, pellets made with 40% RDF-3 have the highest hydrogen (11.76%) and sulfur (1.45%) contents. According to Garcia et al. [33], RDF has a significantly higher content of ash, N, S, and Cl. As a result, it is understandable that pellets made from mixed rubberwood and RDF contain a higher concentration of these constituents.

Table 5 shows the results of the proximate analysis of samples for moisture content (MC), volatile matter (VM), ash content, and fixed carbon (FC). For woody raw materials, moisture content in the range of 5–10% is usually optimal [29]. W5R5 pellets has an excellent moisture content. However, it is recommended that a moisture content of 5–12% be required to produce a high-quality product, because pellets with less than 4% moisture content can absorb moisture from the environment [34]. The average MC content of the samples was 5.50%, compared to 9.61% for wood pellets. It can be concluded that they have the ability to withstand water absorption. Meanwhile, the addition of RDF-3 decreased the volatile matter levels in the pellet samples (Figure 5). The volatile matter content of pure rubberwood pellets is 72.03%. The volatile matter content was reduced to 65.38% and 68.24% after mixing with RDF-3. When RDF-3 was mixed during the pellet production process, however, an increase in ash content was also observed. The increase in ash content was statistically significant and increased with the increasing RDF-3 ratio. On the contrary, as the RDF-3 ratio increased, the fixed carbon content decreased. Garcia et al. [33] made a similar observation, stating that pine pellets have less ash but more fixed carbon than RDF pellets.

Figure 6 depicts the SEM images of the rubberwood pellet and mixed pellet samples. The images show the homogeneous mixing of pellets. Figure 6a exhibits the fiber and texture of rubberwood [35]. Meanwhile, Figure 6b exhibits the texture of rubber wood mixed with RDF-3 compositions via a heating process. The SEM image revealed a smooth surface when compared to that of a rubberwood pellet. Generally, RDF compositions include cardboard, plastic, textile, and organic matter, depending on the weight of the waste [29]. Therefore, the surface of the mixed pellet could form a smooth surface area, and the RFD fabricated the gum to be combined with the wood texture that made stronger RDF–wood pellets on their mechanical durability. Both figures had almost identical surface features, indicating that mixed pellets have the same commercial potential as rubberwood pellets. However, it seems that the RDF-3 samples have a more compacted structure based on their smoother surface, and this characteristic could be a reason for the improved mechanical durability of RDF pellets.

## 4. Conclusions

This study demonstrated that RDF-3, a product of municipal solid-waste transformation, can be mixed with rubberwood to produce a pellet fuel. Pellets with a mixture of 50% rubberwood sawdust and 50% RDF-3 demonstrated the capability of a fuel material with the highest density and energy. The increased RDF ratio had a significant impact on the heating value of mixed pellets, while the volatile matter values were also significantly reduced. Meanwhile, the mechanical durability of the W5R5 sample was 99.07%. Combining both feedstock types appears to have the potential to yield value-added blend pellets for use in biomass power plants and other applications. Other properties, such as the elemental compositions of mixed pellets, would be interesting to investigate in future research to gain a better understanding of how this type of product behaves.

## Figures and Tables

**Figure 1 materials-15-03093-f001:**
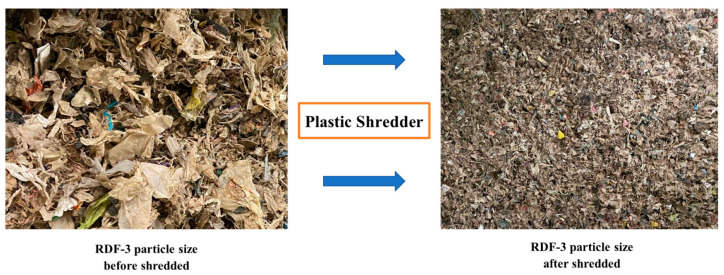
The preparation of the RDF sample.

**Figure 2 materials-15-03093-f002:**
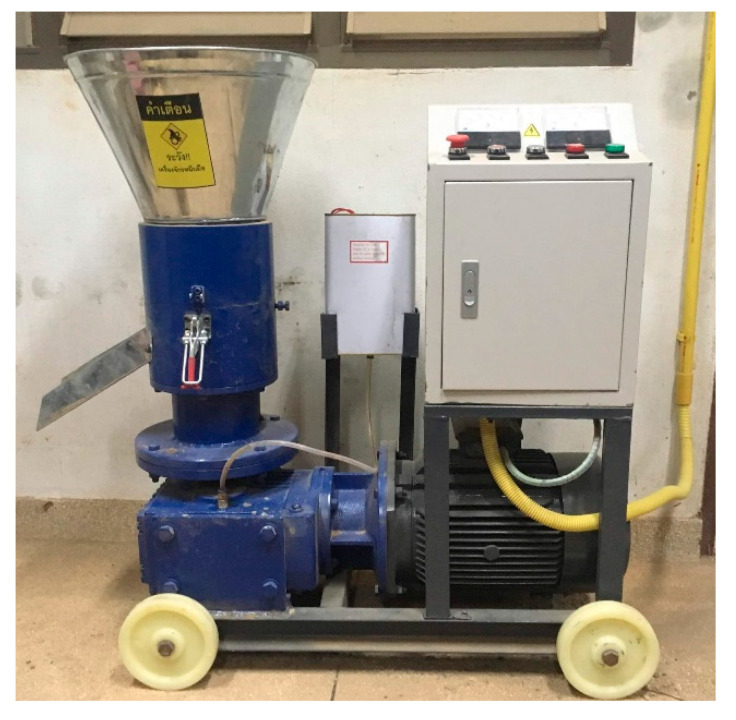
A flat die wood pellet machine for the experiment.

**Figure 3 materials-15-03093-f003:**
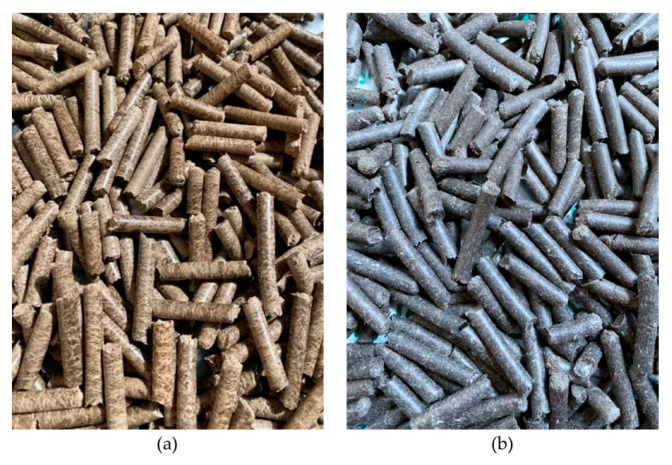
Visual appearance of (**a**) rubberwood pellets and (**b**) mixed pellets, i.e., 50% rubberwood and 50% RDF.

**Figure 4 materials-15-03093-f004:**
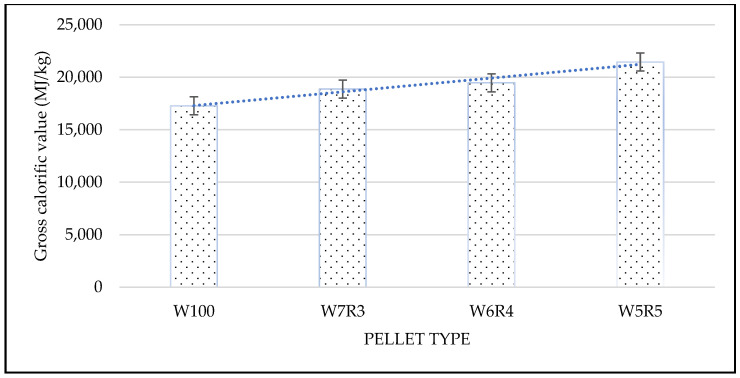
Gross calorific values of pellets made from rubberwood and RDF.

**Figure 5 materials-15-03093-f005:**
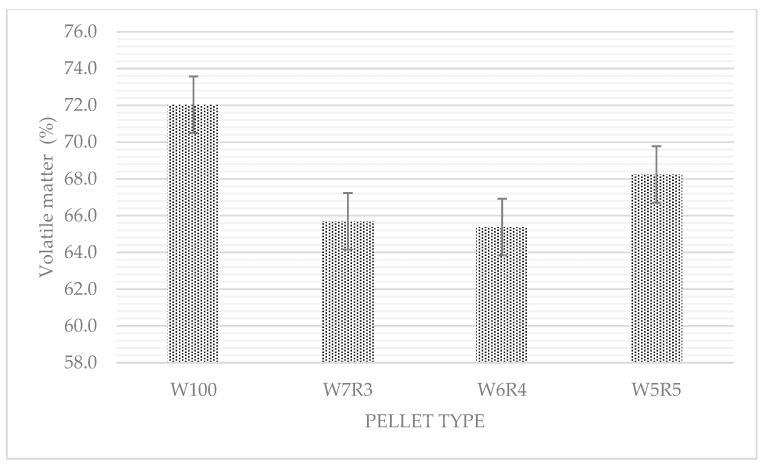
Volatile matter (VM) values of rubberwood and RDF pellets.

**Figure 6 materials-15-03093-f006:**
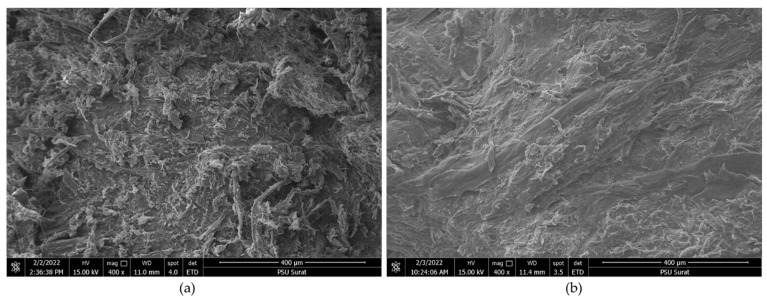
SEM images of (**a**) wood pellet and (**b**) mixed pellet.

**Table 1 materials-15-03093-t001:** The composition of pellet samples.

Sample Type	Rubberwood (%*w*/*w*)	RDF (%*w*/*w*)
W100	100	-
W7R3	70	30
W6R4	60	40
W5R5	50	50

**Table 2 materials-15-03093-t002:** Diameter, length, density, and color appearance of the pellet samples.

Sample	Diameter		Length		Density		Color
Type	(mm)		(mm)		(g/cm^3^)		
	Mean		Mean		Mean		
W100	6.11 ^b^	±(0.02)	36.27 ^c^	±(0.33)	1.288 ^a^	±(0.011)	brown
W7R3	6.19 ^a^	±(0.01)	43.40 ^a^	±(0.29)	1.121 ^d^	±(0.003)	black
W6R4	6.13 ^b^	±(0.05)	41.49 ^b^	±(1.05)	1.175 ^c^	±(0.022)	black
W5R5	6.21 ^a^	±(0.03)	41.64 ^b^	±(0.57)	1.234 ^b^	±(0.018)	black

Numbers in parentheses are standard deviation values. Mean values with the different letters are significantly different at *p* < 0.05.

**Table 3 materials-15-03093-t003:** Mechanical durability and heating values of samples.

Sample	Mechanical		Calorific Value (MJ/kg)	
Type	Durability (%)			
	Mean		Mean	
W100	98.39 ^b^	±(0.32)	17,277 ^d^	±(60)
W7R3	98.27 ^b^	±(0.09)	18,866 ^c^	±(230)
W6R4	98.85 ^a^	±(0.07)	19,461 ^b^	±(83)
W5R5	99.07 ^a^	±(0.09)	21,445 ^a^	±(520)

Numbers in parentheses are standard deviation values. Mean values with the different letters are significantly different, *p* < 0.05.

**Table 4 materials-15-03093-t004:** Ultimate analysis of rubberwood and RDF pellets.

Sample Type	Ultimate Analysis									
	C		H		N		S		Cl	
	Mean (%)		Mean (%)		Mean (%)		Mean (%)		Mean (%)	
W100	47.81 ^b^	±(0.54)	7.74 ^b^	±(1.27)	0.28 ^b^	±(0.05)	0.57 ^d^	±(0.02)	0.017 ^c^	±(0.002)
W7R3	47.59 ^b^	±(0.36)	10.61 ^a^	±(1.19)	0.47 ^b^	±(0.05)	0.90 ^c^	±(0.02)	0.093 ^b^	±(0.005)
W6R4	47.87 ^b^	±(0.22)	11.76 ^a^	±(0.78)	0.47 ^b^	±(0.20)	1.45 ^a^	±(0.13)	0.118 ^a^	±(0.003)
W5R5	50.78 ^a^	±(1.72)	10.35 ^a^	±(0.78)	0.69 ^a^	±(0.03)	1.06 ^b^	±(0.08)	0.124 ^a^	±(0.002)

Numbers in parentheses are standard deviation values. Mean values with the different letters are significantly different, *p* < 0.05.

**Table 5 materials-15-03093-t005:** Proximate analysis of rubberwood and RDF pellets.

Sample Type	Proximate Analysis							
	MC		VM		Ash		FC	
	Mean (%)		Mean (%)		Mean (%)		Mean (%)	
W100	9.61 ^c^	±(0.21)	72.03 ^a^	±(0.03)	2.50 ^d^	±(0.13)	15.85 ^a^	±(0.22)
W7R3	11.17 ^a^	±(0.44)	65.69 ^c^	±(0.43)	9.90 ^c^	±(0.09)	13.24 ^b^	±(0.47)
W6R4	10.49 ^b^	±(0.10)	65.38 ^c^	±(0.10)	11.87 ^b^	±(0.23)	12.26 ^b^	±(0.32)
W5R5	5.50 ^d^	±(0.41)	68.24 ^b^	±(0.76)	13.78 ^a^	±(0.18)	12.48 ^b^	±(0.84)

Numbers in parentheses are standard deviation values. Mean values with the different letters are significantly different *p* < 0.05.

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
