# Peer review of "Characterization of Mixed Pellets Made from Rubberwood (*Hevea brasiliensis*) and Refuse-Derived Fuel (RDF) Waste as Pellet Fuel"

_materials, 2022, doi:10.3390/ma15093093_

Round 1
Reviewer 1 Report
The manuscript presents a study on the co-pelletization of rubberwood and RDF. In general, I found that the novelty of the study is not clear yet. Therefore the authors should improve the manuscript to have a more clarity on the novelty. Also, more discussion is needed on the observed data trend.
- The writing should be improve, a professional proofreading is required.
- I suggest to not put the abbreviation names of the sample (i.e., RDF-3 and W5R5) in the abstract. Instead, the authors can replace it with a full name or specific material content of the samples.
- The introduction is not written in a clear and comprehensive way. Please add more explanation to justify the novelty of the paper, especially for the following aspects,
- there are already a significant number of publications that investigate the co-pelletization of woody biomass and MSW/RDF. Please add a review of theses related studies in the introduction and indicate the gap between the previous study and the authors' study.
- why the author choose rubberwood? Is there any unique features/properties compared to the other woody biomass from other co-pelletization studies?
- the authors should focus more on reviewing the benefit and drawbacks of combining biomass and RDF in term of the mechanical properties of the pellets, not only in term of its utilization as pyrolysis fuel. Also, why the authors only focus on pyrolysis? How about other thermochemical processes?
- Can the authors provide the materials composition of the RDF?
- Why the authors only investigated RDF mixtures lower than 50%?
- Based on the SEM images, authors stated that RDF-3 could improve the mechanical durability of pellets based on its smooth surface of the sample. What is the basis of this statement? Please elaborate more on this matter.
- It is stated in the conclusion section, "The results of mixed pellets containing 50% sawdus and 50% RDF-3 demonstrated the capability of a fuel material with the lowest density but the highest energy." I think it should be the highest density.
Author Response
The manuscript presents a study on the co-pelletization of rubberwood and RDF. In general, I found that the novelty of the study is not clear yet. Therefore the authors should improve the manuscript to have a more clarity on the novelty. Also, more discussion is needed on the observed data trend.
- The manuscript has been improved. The novelty of the study has been mentioned. A more detailed discussion has also been added.
- The writing should be improve, a professional proofreading is required.
- The manuscript was improved some English corrections in the structure of the text by a professional proofreading.
- I suggest to not put the abbreviation names of the sample (i.e., RDF-3 and W5R5) in the abstract. Instead, the authors can replace it with a full name or specific material content of the samples.
- Full name and specific material content of the samples has been used in the abstract.
- The introduction is not written in a clear and comprehensive way. Please add more explanation to justify the novelty of the paper, especially for the following aspects,
- there are already a significant number of publications that investigate the co-pelletization of woody biomass and MSW/RDF. Please add a review of theses related studies in the introduction and indicate the gap between the previous study and the authors' study.
ü The introduction has been improved by including more previous works done on the subject to enhance overall quality of the section.
- why the author choose rubberwood? Is there any unique features/properties compared to the other woody biomass from other co-pelletization studies?
- Rubberwood is the most vital raw material for the wood-based industries in both Thailand and Malaysia (Shigematsu et al. 2011). In comparison to other type of biomass, rubberwood displayed higher potential energy production compared to that of empty fruit bunches and palm kernel shells (Ratnasingam et al. 2015). Comparing to pellets made from oil palm biomass, which is widely available in both Malaysia and Thailand, rubberwood exhibited higher potential in pellet produc-tion owing to its higher calorific value than palm kernel shell and fibre. In addition, pellets made from rubberwood also offers several advantages, such as i) it has better and more uniform heating properties per unit volume; ii) generate fewer particulate emissions during burning; and iii) lower transportation fee due to their increased bulk density (Ratnasingam et al. 2015). This study could improve the value-added utilization of rubberwood. Rubberwood is known for its great potential as a source of energy.
- the authors should focus more on reviewing the benefit and drawbacks of combining biomass and RDF in term of the mechanical properties of the pellets, not only in term of its utilization as pyrolysis fuel. Also, why the authors only focus on pyrolysis? How about other thermochemical processes?
- More previous works has been added. The current study is focused on the pyrolysis where the thermochemical processes will be the focus of the future study.
- Can the authors provide the materials composition of the RDF?
- We apologize for not being able to provide the materials composition of the RDF at the moment as we are lacked of resources.
- Why the authors only investigated RDF mixtures lower than 50%?
- The study used lesser amount of RDF mixtures because the main purpose of this study was emphasized on utilizing more sawdust, which is a waste product from wood industry.
- Based on the SEM images, authors stated that RDF-3 could improve the mechanical durability of pellets based on its smooth surface of the sample. What is the basis of this statement? Please elaborate more on this matter.
ü The clarification based on the SEM images has been added based on your comment.
- It is stated in the conclusion section, "The results of mixed pellets containing 50% sawdus and 50% RDF-3 demonstrated the capability of a fuel material with the lowest density but the highest energy." I think it should be the highest density.
- Thank you for pointing out, the sentence has been revised.

Reviewer 2 Report
The topic of this manuscript is interesting, however there are some issues to be addressed towards the improvement of its quality. Here are my comments to the authors in this direction.
In the abstract it is referred in the same sentence twice the phrase "on average" (remove the one of them). Please, clarify if you refer to lower or higher heating value in the abstract. In the abstract you rather change "of amount" to "compared to", in order to clarify the meaning. Generally, I would recommend to the authors to let a native speaker check and imporve the whole manuscript english language use, since some points need to be clarified and some syntactical/grammatical/type errors to be corrected. The introduction is well approached, but the literature review and state-of-the-art description seem to be a little poor. Please, provide a brief statement about the participation of other lignocellulosic waste materials (except for wood) in pellets and I would recommend you among others to refer to the relevant work of DOI: 10.15376/biores.12.4.9263-9272, to support your statements. In the last paragraphs of introduction you mention "pyrolysis mechanisms of biomass and plastics" without explaining the significant findings of this study. Additionally, the phrase "revealed an understanding" needs to be improved.
Please provide information on the raw material of rubberwood sawdust (number of trees/where was this sawdust was obtained from, which part of the tree, since the chemical composition is being changed etc.). Why did you dry the material of rubberwood to 0%? (since the presence of moisture is beneficial to the consistency and durability of the produced pellets). Given that the wood is highly hygroscopic material, is it possible to mix it in this situation (totally dry)? Did you sieve your material in order to investigate the ratio of different particles size classes? Please explain. In figure 1, "before shredded" and "after shredded" need improvement. which was the exact moisture content of the 2 materials during mixing and pelletizing process? You should refer all the standards used in properties evaluation. Please, provide the calorific value in SI units (MJ/kg) and not kcal/kg. You did not mention the methodology for volatile organic compounds (VOCs), ash content (AC), and fixed carbon (FC) investigation. Provide some details on proximity analysis to be clarified to the reader. Please, provide also in the text a figure of the flat die pellet mill. I believe that the colour of pellets is influenced by numerous factors, so how could the colour measurement results be utilized in this study? In figure 3, clarify that you refer to gross calorific values. Some missing doi numbers to be provided.
Author Response
The topic of this manuscript is interesting, however there are some issues to be addressed towards the improvement of its quality. Here are my comments to the authors in this direction.
In the abstract it is referred in the same sentence twice the phrase "on average" (remove the one of them). Please, clarify if you refer to lower or higher heating value in the abstract. In the abstract you rather change "of amount" to "compared to", in order to clarify the meaning.
- The abstract was revised following your comments.
Generally, I would recommend to the authors to let a native speaker check and imporve the whole manuscript english language use, since some points need to be clarified and some syntactical/grammatical/type errors to be corrected.
- As the reviewer suggested by a native speaker checking on the manuscript, it was improved some English corrections in the structure of the text by a professional proofreading.
The introduction is well approached, but the literature review and state-of-the-art description seem to be a little poor. Please, provide a brief statement about the participation of other lignocellulosic waste materials (except for wood) in pellets and I would recommend you among others to refer to the relevant work of DOI: 10.15376/biores.12.4.9263-9272, to support your statements. In the last paragraphs of introduction you mention "pyrolysis mechanisms of biomass and plastics" without explaining the significant findings of this study. Additionally, the phrase "revealed an understanding" needs to be improved.
- The introduction has been improved by including more previous works done on the subject to enhance overall quality of the section.
Please provide information on the raw material of rubberwood sawdust (number of trees/where was this sawdust was obtained from, which part of the tree, since the chemical composition is being changed etc.).
- The information of the raw material of rubberwood sawdust was provided in line 105-106.
Why did you dry the material of rubberwood to 0%? (since the presence of moisture is beneficial to the consistency and durability of the produced pellets). Given that the wood is highly hygroscopic material, is it possible to mix it in this situation (totally dry)?
- We need to control all material set at 0% MC then we added moisture content to mixed material until 14-16% MC by distilled water before pelletization. It is the best MC that can product pellets firmly.
Did you sieve your material in order to investigate the ratio of different particles size classes? Please explain.
- The experiment was sieved only oversize of sawdust by using sieve No. 5. There is no investigation on the ratio of different particles size classes.
In figure 1, "before shredded" and "after shredded" need improvement. which was the exact moisture content of the 2 materials during mixing and pelletizing process? You should refer all the standards used in properties evaluation.
- In Figure 1, "before shredded" and "after shredded" was edited.
Please, provide the calorific value in SI units (MJ/kg) and not kcal/kg.
üThe data of pellets was provided the calorific value in SI units (MJ/kg).
You did not mention the methodology for volatile organic compounds (VOCs), ash content (AC), and fixed carbon (FC) investigation.
- The methodology for volatile matter (VM), ash content (AC), and fixed carbon (FC) were provided.
Provide some details on proximity analysis to be clarified to the reader.
- Proximity analysis has been provided in Table 5 and some discussions have been included in the text.
Please, provide also in the text a figure of the flat die pellet mill. I believe that the colour of pellets is influenced by numerous factors, so how could the colour measurement results be utilized in this study?
- A Figure of the flat die pellet mill was added as in Figure 2. The colour pellets presented are for illustrating purpose to show the different colour between rubberwood pellets and rubberwood-RDF pellets.
In figure 3, clarify that you refer to gross calorific values.
- Gross calorific values have been mentioned in the title of the Figure (now Figure 4).
Some Some missing DOI numbers were provided in references.
- Missing doi numbers has been provided in the references list.

Reviewer 3 Report
The study presented by the authors is very interesting as it outlines the characterisation of pellets manufactured from RDF and rubberwood, a commonly found raw material in the authors' geographical context. The techniques used for the said characterisation have been well outlined and discussed. Finally, the discussion part of the paper is clear and the conclusions drawn are logical.
However, I found the introduction part to be very broad. It does not include any previous work done on the subject so that the authors can compare their results. There are some studies quoted, but they need to be more detailed. Apart from this and some English corrections in the structure of the text (as can be found in the attached file), I have no further comments."
Thank you for your email. I hope these comments are satisfactory. I look forward to working with you again soon.

Author Response
The study presented by the authors is very interesting as it outlines the characterisation of pellets manufactured from RDF and rubberwood, a commonly found raw material in the authors' geographical context. The techniques used for the said characterisation have been well outlined and discussed. Finally, the discussion part of the paper is clear and the conclusions drawn are logical.
However, I found the introduction part to be very broad. It does not include any previous work done on the subject so that the authors can compare their results. There are some studies quoted, but they need to be more detailed.
- The introduction has been improved by including more previous works done on the subject to enhance overall quality of the section.
Apart from this and some English corrections in the structure of the text (as can be found in the attached file), I have no further comments."
- Some English corrections in the structure of the text have been revised based on your attached file.

Round 2
Reviewer 1 Report
The authors' have provided sufficient answers according to the following comments. Nevertheless, I feel that the discussion on the mechanical durability can be more comprehensive. I would suggest the authors to add a brief explanation on the effects of plastic content on the RDF-wood pellet in the final manuscript. The presence of plastic can be the caused of the stronger RDF-wood. This is also perhaps can explain the difference on the SEM profiles.
Author Response
Response to Reviewer 1 Comments
Response to reviewers (materials-1670241)
Reviewer 1 (Round 2)
The authors' have provided sufficient answers according to the following comments. Nevertheless, I feel that the discussion on the mechanical durability can be more comprehensive. I would suggest the authors to add a brief explanation on the effects of plastic content on the RDF-wood pellet in the final manuscript. The presence of plastic can be the caused of the stronger RDF-wood. This is also perhaps can explain the difference on the SEM profiles.
- The manuscript has been improved. The explanation was provided in line 269-271.

Reviewer 2 Report
As I have checked, the authors have implemented the proposed changes in the revised verion of manuscript towards the improvement of their work. Please make also the following correction of reference 17 in bibliography list:
Kamperidou V., Lykidis C., Barmpoutis P., 2017. Assessment of the Thermal Characteristics of Pellets Made of Agricultural Crop Residues Mixed with Wood. BioResources 12 (4): 9263-9272. DOI: 10.15376/biores.12.4.9263-9272.
Author Response
Response to Reviewer 2 Comments
Response to reviewers (materials-1670241)
Reviewer 2 (Round 2)
As I have checked, the authors have implemented the proposed changes in the revised version of manuscript towards the improvement of their work. Please make also the following correction of reference 17 in bibliography list:
- A reference has been revised correctly in line 339.
